# Factors Associated with the Implementation of an Improved Community Health Fund in the Ubungo Municipality Area, Dar es Salaam Region, Tanzania

**DOI:** 10.3390/ijerph19095606

**Published:** 2022-05-05

**Authors:** Goodluck Mselle, Peter Nsanya, Kennedy Diema Konlan, Yuri Lee, Jongsoo Ryu, Sunjoo Kang

**Affiliations:** 1Department of Health and Social Welfare, Ubungo Municipal Council, Dar es Salaam P.O. Box 55068, Tanzania; gmselle10@gmail.com (G.M.); nsanyap@gmail.com (P.N.); 2Department of Public Health Nursing, School of Nursing and Midwifery, University of Health and Allied Sciences, Ho, Ghana; dkkonlan@uhas.edu.gh; 3Mo-Im Kim Nursing Research Institute, College of Nursing, Yonsei University, Seoul 03722, Korea; 4Department of Health and Medical Information, Myongji College, Seoul 03674, Korea; wittyyurilee@gmail.com; 5Department of Global Health, Graduate School of Public Health, Yonsei University, Seoul 03722, Korea; johnryuhira@gmail.com

**Keywords:** Improved Community Health Fund (iCHF), Ubungo municipality, healthcare providers, community health, health insurance

## Abstract

Community-based health insurance schemes help households to afford healthcare services. This paper describes healthcare facilities and community factors that are associated with the Improved Community Health Fund (iCHF) scheme in the Ubungo district of Tanzania. A cross-sectional descriptive study was conducted using online questionnaires that were completed by healthcare providers and community members in public-owned healthcare facilities in the Ubungo Municipal Council district of Dar es Salaam, Tanzania, between October and November 2021. The data were analyzed using descriptive statistics and the chi-squared test of association. We found a statistically significant relationship between income level and satisfaction with the iCHF scheme. For community-related factors, income level was statistically significant in the level of involvement in iCHF implementation among local leaders. Further, income level was statistically significant in relation to community behavior/culture toward the iCHF. Occupation was statistically significant in iCHF implementation, iCHF premiums, and iCHF membership size. A statistically significant relationship was also found between income, iCHF membership size, and iCHF premiums. Moreover, people would be willing to pay the required premiums if the quality of the healthcare services under the iCHF scheme improves. Therefore, the government should allocate resources to reduce the challenges that are facing iCHF implementation, such as the preference for a user fee scheme over the iCHF, the issues that are faced by enrollment officers, and inadequate iCHF premiums and membership size.

## 1. Introduction

Health finance in low-income countries (LIC) is derived from various sources, including government allocations, out-of-pocket payments, and external aid from high-income countries [1,2]. Out-of-pocket payments are direct payments that are made by citizens for the use of healthcare services. The burden of out-of-pocket payments varies depending on household size, the nature of any relevant health insurance cover, and the presence of chronic diseases in the family [3]. Out-of-pocket healthcare expenditure is large in low-income countries and poses a major barrier for community access to healthcare services. In 2018, out-of-pocket healthcare expenditure accounted for 42% of the total government healthcare budget [1], whereas social health insurance accounted for only 7% of all healthcare-related expenditure [1]. In contrast, for high-income countries, social health insurance funds accounted for 22% of their budget and only 21% was accounted for by out-of-pocket expenditure [1]. In the Universal Health 525 Coverage (UHC) report by the World Health Organization (WHO), out-of-pocket expenditure in Africa was primarily spent on medicines and medical supplies [2]. Moreover, it exceeded the 25% limit that was set by the UHC for the Sustainable Development Goal (SDG) indicator for out-of-pocket expenditure [4]. As per the Abuja Declaration, each nation must spend 15% of its total budget on healthcare, yet Tanzania only spent 8% of its budget on healthcare [4]. Tanzania also only spent USD 46 per capita on healthcare and 7.3% of the GDP on healthcare goods and services in 2014–2015, compared to 10% in Rwanda [5]. Out-of-pocket payments accounted for 26% of the total healthcare expenditure in Tanzania between 2014 and 2015. Tanzania is struggling to find the resources to create a robust healthcare budget and depends on external sources, which account for 37% of the country’s total healthcare budget [6].

The Improved Community Health Fund (iCHF) is a community-based voluntary insurance program that was established in 2001 in Tanzania, which targets the populations that cannot access the National Health Insurance Fund (NHIF). In Tanzania, the NHIF primarily caters for the healthcare needs of government and private employees, who constitute only 7% of the population [7]. The NHIF covers families with up to eight children who are enrolled at dispensaries, health centers, and district hospitals for medical costs of between USD 12 and USD 65 per year. In addition, members of the NHIF benefit from outpatient care and referral systems to district or zonal hospitals. The central government, local government, and iCHF members also contribute to a matching grant [2]. The Community Health Fund (CHF) is an alternative healthcare financing measure for families who are not qualified for or cannot afford the NHIF. In 2015, the government of Tanzania set the target of enrolling 30% of the population under the CHF; however, only 16.4% were enrolled. In 2016, the government had to revise the CHF in order to increase enrollment and improve the quality of the healthcare services that were provided under the scheme. The new changes pertained to enrollment, premium amounts, collection mechanisms, and purchasing arrangements. The new version of the CHF was named the iCHF [8]. In 2018, iCHF coverage increased to 25%, with 13 million beneficiaries [7], although the number of beneficiaries was still limited according to the 2015 government projections [7]. In a study of the Mtwara region in Tanzania, healthcare facility factors, such as the absence of medicines and medical supplies, were found to cause dropouts from iCHF membership [9]. Additional challenges, such as the poor supervision of healthcare facilities by district health authorities and the mismanagement of iCHF funds, led to an insufficient enrollment of CHF members [10]. Therefore, iCHF implementation in Tanzania has faced many challenges. 

To address the urgent need to examine the factors that are associated with the implementation of the iCHF, the present study of the Ubungo municipality in the Dar es Salaam region of Tanzania, where iCHF coverage is low, aimed to identify those factors. To address the factors that influence the implementation of the iCHF, specific focus needs to relate to the healthcare providers and community members who are involved in the service implementation and utilization, respectively. Therefore, this study aimed to describe the community and healthcare facility factors that influence the implementation of the iCHF in Tanzania.

## 2. Materials and Methods

### 2.1. Study Design and Setting

Using a descriptive cross-sectional design, data were collected from healthcare facilities that are registered under the iCHF. Healthcare providers who were working in a total of 21 public-owned healthcare facilities (17 dispensaries, 3 health centers, and 1 district hospital) in the Ubungo municipality were identified for the study. The Ubungo district is an urban area with a population of 1,043,549 people. The district covers a total surface area of 210 square kilometers and has 91 streets (the lowest level administrative area) and 14 wards (the mid-level administrative area) [11].

### 2.2. Population and Sampling

A multistage sample method was adopted for the selection of the facilities and respondents for this study. A total of 13 iCHF members were selected from each facility to participate in the study, using the simple random sample method. In using this sampling method, a sample framework of the iCHF members within each facility was first created and listed in a basket. Names were then drawn from the basket without any replacements. Those whose names were selected through this process were then contacted to respond to a questionnaire. A simple random sampling method was used to select 16 healthcare facilities out of the 21 public healthcare facilities that are under the Ubungo Municipal Council [12]. The sampling of the healthcare facilities was also performed by firstly creating a sample framework of the 21 healthcare facilities and then facilities were chosen without replacement until 16 facilities were selected. The respondents were required to provide consent prior to responding to the questionnaire. During the collection, 29 people did not return the questionnaire or did not initially provide consent to participate in the study. This represented a non-response rate of 13.9%. During data cleaning, five questionnaires were found to be completed inappropriately and hence, were also excluded from the analysis. Therefore, the effective sample size that was used for analysis was 174. 

### 2.3. Measurements

The design and content of the online questionnaires were obtained from the existing literature on the factors that affect community-based health insurance (CBHI) [13]. The questionnaire for the healthcare providers had three parts: questions regarding the demographic characteristics of the participants, closed-ended questions regarding the healthcare facility and community-related factors (independent variables), and questions on iCHF implementation The closed-ended questions were measured using a 5-point Likert-type scale (5 = strongly agree, 4 = agree, 3 = neutral, 2 = disagree, and 1 = strongly disagree). The demographic factors that were assessed for healthcare providers were level of healthcare facility, age, sex, profession, and educational level, while those for community members were age, sex, marital status, family size, occupation, educational level, income level, and duration of iCHF enrollment. The questionnaire for the healthcare providers also elicited their rating of facility factors that influence the implementation of the iCHF on a scale of agree, neutral or disagree. Some of the healthcare facility factors that they rated included the availability of medical equipment and healthcare personnel, the effectiveness of the referral systems, and the awareness of promotion strategies. Community members were also asked to provide similar ratings on their level of satisfaction of the implementation of the iCHF, the involvement of community members in membership registration, and the influence of cultural factors on the implementation of the iCHF. Variables that were rated by both the community members and healthcare professionals included difficulty in paying iCHF premiums, membership size, and the availability of enrolment officers.

Before data collection, the questionnaire was pretested among healthcare providers and community members in the Ubungo district. The questions were uploaded via Google Forms and distributed to 20 healthcare providers at the district hospital and dispensaries and 10 community members. The questionnaire was reviewed by the Ubungo Municipal Council’s Health Department Research Review Committee. The Cronbach’s alpha was 0.785 for the community questionnaire and 0.707 for the healthcare provider questionnaire.

### 2.4. Data Analysis

The data analysis was performed using the SPSS 25.0 software after downloading the data from Google Forms into Excel spreadsheets. A descriptive analysis of the sociodemographic data was conducted using frequencies, means (M), and standard deviations (SD). A chi-squared test was performed to measure the association between sociodemographic factors and iCHF implementation. The statistical significance was set at *p* < 0.05 for a 95% confidence interval.

## 3. Results

### 3.1. Sociodemographic Characteristics

Among the participants who were healthcare providers, 75.9% worked in healthcare centers/dispensaries, 55.8% were below 39 years of age, and 70.1% were female (Table 1). 

Among the participants from the community, 79.4% were aged over 29 years, 52.8% were male, and 77.4% were married. The main occupations of the community members were office work (55.8%) and business (44.2%). As shown in Table 2, additional sociodemographic factors that were reported by the community members included education level, income level, and health insurance enrollment status.

### 3.2. Healthcare Facility-Related Characteristics That Affect the iCHF

The findings of the present study showed that the level of the healthcare facility and the adequate availability of healthcare providers were statistically significant (χ^2^ = 15.70, *p* < 0.00). A total of 62.1% of the healthcare workers who were working in health centers and dispensaries agreed that there were enough workers in the healthcare facilities, as shown in Table 3. The educational level of the healthcare professionals was also statistically significant (χ^2^ = 12.56, *p* < 0.00) with an efficient referral system under the iCHF scheme. A total of 88.7% of the nurses and other workers agreed that the referral system was working well under the iCHF scheme. The relationship between education level and the availability of healthcare providers in a healthcare facility was statistically significant (χ^2^ = 38.99, *p* = 0.00). A total of 55% of the community members with university-level education agreed that there were enough healthcare providers in the healthcare facilities. The relationship between income level and iCHF awareness and promotion strategies in the community was statistically significant (χ^2^ = 17.60, *p* < 0.00). A total of 83% of the community members from low-income groups agreed that iCHF awareness and promotion strategies were conducted in the community. The relationship between occupation and the availability of medical supplies/equipment in a healthcare facility was statistically significant (χ^2^ = 51.68, *p* < 0.00), as 81.5% of the community members who were running businesses agreed that there were enough medical supplies/equipment in the healthcare facilities. Additionally, there was a statistically significant (χ^2^ = 43.59, *p* < 0.00) relationship between occupation and the availability of healthcare providers in a health facility. A total of 81.8% of the community members who were running businesses or were self-employed agreed that there were enough healthcare providers in the healthcare facilities. A statistically significant (χ^2^ = 52.86, *p* < 0.00) relationship was also identified between occupation and an efficient referral system under the iCHF scheme, as 76% of the community members who were running businesses agreed that the referral system was working well. The relationship between occupation and the awareness and promotion strategies for the iCHF in the community was also statistically significant (χ^2^ = 37.91, *p* = 0.000), as 47.8% of the community members who were running businesses agreed that awareness and promotion strategies for the iCHF were conducted in the community. 

A statistically significant (χ^2^ = 46.02, *p* < 0.00) relationship was found between education level and the involvement of local leaders, as shown in Table 3. A total of 59% of the community members with secondary- or primary-level education agreed that community leaders were involved in iCHF implementation. In addition, the relationship between education level and community behavior/culture toward the iCHF was statistically significant (χ^2^ = 16.65, *p* < 0.00), as 41% of the community members with secondary- or primary-level education agreed that members of the community seek to register with the iCHF after they have had an illness (Table 3). A statistically significant (χ^2^ = 12.07, *p* < 0.000) relationship was also found between income level and the involvement of local leaders in iCHF implementation. A total of 85% of the community members with low income levels agreed that local leaders participated in iCHF implementation. Moreover, the relationship between income level and community behavior/culture toward the iCHF was statistically significant (χ^2^ = 16.92, *p* < 0.00). A total of 72% of the community members with low income levels agreed that members of the community seek to register with the iCHF after an illness.

### 3.3. Institution-Related Factors That Affect the iCHF 

A statistically significant (χ^2^ = 14.01, *p* = 0.001) relationship was identified between education level and ability to pay the iCHF premiums. A total of 50% of the community members with university-level education agreed that the iCHF premiums were affordable. The relationship between education level and iCHF membership size was also statistically significant (χ^2^ = 14.01, *p* < 0.00). A total of 46% (χ^2^ = 14.01, *p* < 0.001) of the community members with university-level education disagreed that iCHF membership should require the payment of premiums (Table 4). The relationship between income level and iCHF membership size was statistically significant (χ^2^ = 21.818, *p* = 0.000) as well. A total of 75% of the community members with low income levels disagreed regarding iCHF membership size. Finally, the relationship between income level and ability to pay the iCHF premiums was statistically significant (χ^2^ = 18.68, *p* = 0.00). A total of 68% of the community members with low income levels agreed that the iCHF premiums were affordable. A statistically significant (χ^2^ = 19.08, *p* < 0.001) relationship was found between occupation and ability to pay the iCHF premiums. A total of 61.4% of the community members who were running businesses agreed that the iCHF premiums were affordable. The relationship between occupation and iCHF membership size was statistically significant (χ^2^ = 21.289, *p* < 0.00) as well. A total of 65.9% of the community members who were running businesses disagreed that iCHF membership was acceptable.

## 4. Discussion

The present study identified the factors that influenced the implementation of the Improved Community Health Fund project in the Ubungo municipality of Dar es Salaam, Tanzania. The results of this study confirmed the availability of sufficient healthcare providers at various levels of healthcare facilities, thereby demonstrating the increased healthcare service provision under the iCHF scheme. The role of healthcare providers was found to be essential for the quality of the services that were provided. Therefore, it is pertinent that government interventions are aimed at improving healthcare service provision in communities by enhancing the quantity and quality of healthcare workers. This would also help to increase the confidence of community members in adopting insurance schemes, such as the iCHF. To ensure this occurs, the government should invest more financial resources into improvements in healthcare service provision, which would automatically increase the coverage and enrollment of iCHF members. Our findings regarding the importance of the availability of qualified healthcare providers were consistent with studies that were conducted in the Iramba and Liwale districts of Tanzania, where the development of healthcare providers influenced the excellent performance of the iCHF scheme [14]. 

Additionally, the necessary medical equipment/supplies were considered to be essential for healthcare and iCHF provision, as members would receive better healthcare services under the scheme. This finding was consistent with a study that was conducted in Ethiopia, where the presence of medical equipment led to an increase in the enrollment of iCHF members [15]. With the increasing quantity and quality of equipment and personnel within healthcare facilities, community confidence in orthodox medicine is likely to improve, along with an increase in enrollment in health insurance schemes. This is particularly true for resource-strapped African countries, in which the use of traditional alternative medicine and treatment has been a barrier to service acquisition. Most alternative medical practices in poor countries lack scientific rigor and are conducted in unsanitary and unsafe environments. This study demonstrated that other factors that affect the implementation of the iCHF included insufficient supplies and a lack of trained professionals. The iCHF would perform better if healthcare facilities had sufficient medical supplies/equipment and healthcare providers. In Ethiopia, the community perceived a poor quality of healthcare under the community-based health insurance scheme due to the unavailability of medicines/medical supplies, long wait times, and inadequate healthcare providers [13]. In Tanzania, poor healthcare outcomes under the iCHF were caused by a lack of healthcare providers, long wait times, and a lack of medications [16,17].

This study indicated that the referral systems under the iCHF were functioning well in the healthcare facilities that were included in the research. This could be due to the proximity of the hospitals in the municipality, the availability of ambulances, and the presence of several referral hospitals and a national hospital in the area. This finding contrasted with many studies that were conducted in other parts of Tanzania, where the referral systems under the iCHF have been reported to be not working well. This could be caused by the absence of referral or zonal hospitals in those areas and the limited number of ambulances, especially in rural areas [18,19]. Healthcare referral systems are essential for improving community perceptions of the benefits that are associated with insurance facilities in low-resource settings. 

Community awareness and promotion strategies contributed to the implementation of the iCHF at the community level by increasing iCHF enrollment [20]. This finding was consistent with studies that were conducted in three Dodoma districts in Tanzania, which showed higher levels of community awareness [21,22]. However, the factors that influence enrollment in community-based health insurance schemes are numerous and interact in complex ways to influence individual decisions. In a study that was conducted in northwest Ethiopia, community awareness programs about saving and credit decisions contributed significantly to the implementation of the community health insurance scheme [21]. In another study in southern Ethiopia, community leaders played a crucial role in enrolling and implementing community-based health insurance [13]. A study in Tanzania indicated that most of the information on iCHF implementation was obtained from local leaders [21]. Another study in Dodoma, Tanzania, indicated that the implementation of the iCHF scheme improved after the involvement of community leaders. District authorities ensured that all local leaders were registered with the iCHF scheme and were informed as to its importance for the local community [23]. Therefore, in addition to an increased awareness of the health insurance scheme, other factors and barriers that influence enrollment need to be further investigated in future studies.

The income and education levels of the community members were important factors that influenced the adoption of iCHF, as the participants with high income and education levels perceived the healthcare services under the iCHF scheme to be of good quality, regardless of the iCHF premiums or iCHF membership size. The results of studies in northwest [24,25] and southwest Ethiopia [26] were consistent with the current findings that the iCHF premiums were acceptable to the community if the scheme could ensure quality healthcare services. However, a study in Rwanda opposed this finding because it was found that a change in the premiums could cause low enrollment rates, especially in rural areas [27]. In addition, high premium costs could contribute to differences in healthcare provision between urban and rural areas due to varying income levels. The results from a previous study in Tanzania were consistent with the current findings that the low levels of iCHF membership in three districts were caused by poor healthcare services and low enrollment. However, in another study, the iCHF premiums were influenced by the user fee and people who were not members of the CHF scheme preferred it because the quality of the healthcare services was good [27].

### Strengths and Limitations

The present study was successful in identifying factors that affect the implementation of the iCHF at both the community member and healthcare provider levels, unlike many of the other studies, which have focused on the factors that affect community health while ignoring the role of healthcare providers. However, a key limitation of this study was the inability to determine the causal relationship between the factors that impact iCHF adoption due to the cross-sectional nature of the study. The factors that were associated with the implementation of the iCHF were multiple dimensional and the level of impact of each variable could only be ascertained through longitudinal studies or those that implement experimental designs. Due to this, we could not determine the level of impact of each factor on the implementation of the iCHF in Tanzania. Therefore, it is important that experimental research designs are employed to test the contribution and efficacy of each influencing factor in terms of an individual’s choice to use the iCHF. However, regardless of these limitations, we highlighted the factors that are associated with the implementation of the iCHF in Tanzania.

## 5. Conclusions

This study demonstrated that multiple dimensional factors influence the implementation of the iCHF in Tanzania. These factors range from individual to community factors, which interact dynamically to influence the implementation of the iCHF. For healthcare providers, the factors that affect the implementation of the iCHF scheme that were identified in the present study included the level of the healthcare facility, the education level of the healthcare providers, and the referral system in the hospital. For community members, the factors included education level, age, and income level. All health, community, and institutional factors were affected at both the healthcare facility and community levels. It is important that future research uses experimental designs or longitudinal follow-up approaches to identify the level of influence of each factor on the implementation of the iCHF. To have an effective iCHF, stakeholders must focus on mitigating these factors while efforts are made to improve the education and awareness of the tenants of the fund. The education of community members about the iCHF must focus on those in the lower socioeconomic groups, which have limited formal education. Healthcare institutions must also be empowered to implement effective referral systems that can ensure prompt care for patients.

## Figures and Tables

**Table 1 ijerph-19-05606-t001:** Sociodemographic characteristics of healthcare providers.

Items	Categories	Frequency	%	Mean ± SD
Level of Healthcare Facility	District Hospital	42	24.1	
	Health Centers/Dispensaries	132	75.9	
Age	18–39	97	55.8	2.4 ± 0.7
	40–61	77	44.2	
Sex	Male	52	29.9	
	Female	122	70.1	
Profession	Doctor	77	44.3	
	Nurse and Other Workers	97	55.7	
Education Level	Certificate	34	19.7	
	Diploma	103	59.5	

**Table 2 ijerph-19-05606-t002:** Sociodemographic characteristics of community members.

Item	Categories	Frequency	%	Mean ± SD
Age	18–28	41	20.6	2.1 ± 0.7
	29–50	158	79.4	
Sex	Male	105	52.8	
	Female	94	47.2	
Marital Status	Single	45	22.7	
	Married	153	77.4	
Family Size	1–4	118	59.3	1.5 ± 0.63
	5–8	81	40.7	
Occupation	Office Work	111	55.8	
	Business	88	44.2	
Education Level	University/College	135	67.8	
	Primary/Secondary	64	32.2	
Income Level	High	5	2.5	
	Low	194	97.5	
Enrolled Member of iCHF	1–4	174	87.4	1.1 ± 0.4
	5–8	24	12.1	

**Table 3 ijerph-19-05606-t003:** Healthcare facility-related factors that affect the iCHF scheme, according to healthcare providers and community members.

Healthcare Facility-Related Factors							
	Availability of Medical Supplies/Equipment						
Factor	Categories	Agree	Neutral	Disagree	χ^2^	Cramer’s V	Sig.
Education Level	Master’s	51 (37.8)	40 (29.6)	44 (32.6)	44.51	0.473	0.000
	Bachelor’s	14 (38.9)	5 (13.9)	17 (47.2)			
	Diploma	100 (73.0)	15 (10.9)	22 (16.1)			
	Primary/Secondary	56 (87.5)	1 (1.6)	7 (10.9)			
Income Level	High	24 (33.3)	23 (31.9)	25 (34.7)	19.45	0.313	0.000
	Low	83 (65.4)	18 (14.2)	26 (20.5)			
Occupation	Employed	35 (59.7)	37 (33.3)	39 (35.1)	51.68	0.510	0.000
	Self-employed	72 (81.8)	4 (4.5)	12 (13.6)			
Availability of Healthcare Providers in Healthcare Facilities							
Factor	Categories	Agree	Neutral	Disagree	χ^2^	Cramer’s V	Sig.
Level of Healthcare Facility	District Hospital	16 (23.7)	0 (0.0)	26 (61.9)	15.70	0.300	0.000
	Health Centers/Dispensary	82 (62.1)	11 (8.3)	39 (29.5)			
Education Level	University	55 (40.7)	37 (27.4)	43 (31.9)	38.99	0.443	0.000
	Primary/Secondary	56 (87.5)	2 (3.1)	6 (9.4)			
Occupation	Employed	39 (35.1)	33 (29.7)	39 (35.1)	43.59	0.468	0.000
	Self-employed	72 (81.8)	6 (6.8)	10 (11.4)			
Efficiency of the Referral System under the iCHF Scheme							
Factor	Categories	Agree	Neutral	Disagree	χ^2^	Cramer’s V	Sig.
Healthcare Professionals	Doctors	62 (80.5)	4 (5.2)	14 (13.0)	12.56	0.270	0.002
	Nurses and Other Workers	86 (88.7)	10 (10.3)	1 (1.0)			
Education Level	University	59 (43.7)	45 (33.3)	31 (23)	36.99	0.431	0.000
	Primary/Secondary	57 (89.1)	3 (4.7)	4 (6.3)			
Income Level	High	26 (36.1)	30 (41.7)	16 (22.2)	25.29	0.357	0.000
	Low	90 (70.9)	18 (14.2)	19 (15)			
Occupation	Employed	40 (36)	44 (39.6)	27 (24.3)	52.86	0.515	0.000
	Self-employed	76 (51.3)	4 (21.2)	8 (9.1)			
Awareness and Promotion Strategies for the iCHF Scheme							
Factor	Categories	Agree	Neutral	Disagree	χ^2^	Cramer’s V	Sig.
Education Level	University	55 (40.7)	27 (20.0)	53 (39.3)	31.49	0.398	0.000
	Primary/Secondary	53 (82.8)	2 (3.1)	9 (14.1)			
Income Level	High	25 (34.7)	16 (22.2)	31 (43.1)	17.60	0.297	0.000
	Low	83 (65.4)	13 (10.2)	31 (24.4)			
Age	18–28	23 (56.1)	9.8 (4.0)	14 (34.1)	15.95	0.2	0.003
	29–39	46 (44.2)	16 (15.4)	42 (40.4)			
	40–90	39 (72.2)	9 (16.7)	6 (11.1)			
Occupation	Employed	39 (35.1)	25 (22.5)	47 (42.3)	37.90	0.436	0.000
	Self-employed	69 (47.8)	4 (4.5)	15 (17.0)			
Satisfaction with the iCHF Scheme at Healthcare Facilities							
Factor	Categories	Agree	Neutral	Disagree	χ^2^	Cramer’s V	Sig.
Education Level	University	65 (48.1)	23 (17)	47 (34.8)	28.80	0.38	0.000
	Bachelor’s	22 (61)	2 (1.9)	12 (33.3)			
	Diploma	125 (91.2)	7 (5.16)	5 (3.6)			
	Primary/Secondary	56 (38.9)	1 (1.6)	7 (10.9)			
Income Level	High	32 (43.8)	15 (20.8)	25 (34.7)	14.55	0.27	0.001
	Low	89 (70.1)	9 (7.1)	29 (22.8)			
Occupation	Employed	49 (44.1)	23 (20.7)	39 (35.1)	32.98	0.407	0.000
	Self-employed	72 (5.2)	1 (1.1)	15 (23.9)			
Community-Related Factors							
Involvement of Local Leaders in the iCHF Scheme							
Factor	Categories	Agree	Neutral	Disagree	χ^2^	Cramer’s V	Sig.
Education Level	University	56 (41.5)	34 (25.2)	45 (33.3)	46.02	0.481	0.000
	Primary/Secondary	59 (92.2)	1 (1.6)	4 (6.3)			
Income Level	High	30 (41.7)	18 (25)	24 (33.3)	12.07	0.246	0.000
	Low	85 (66.9)	17 (13.4)	25 (19.7)			
Age	18–28	27 (65.9)	3 (7.3)	11 (26.8)	17.94	0.212	0.001
	29–39	48 (46.2)	22 (21.2)	34 (32.7)			
	40–90	40 (74.1)	10 (18.5)	4 (11.1)			
Occupation	Employed	40 (64.1)	30 (19.5)	41 (36.9)	48.72	0.495	0.000
	Self-employed	75 (85.2)	5 (5.7)	8 (9.1)			
Community Behavior/Culture Toward the iCHF Scheme							
Factor	Categories	Agree	Neutral	Disagree	χ^2^	Cramer’s V	Sig.
Education Level	University	51 (37.8)	18 (13.3)	66 (48.9)	16.65	0.289	0.000
	Primary/Secondary	41 (64.1)	0 (0.0)	23 (35.9)			
Income Level	High	20 (33.3)	6.5 (15.3)	41 (56.9)	16.92	0.292	0.000
	Low	72 (58.7)	7 (11.5)	48 (37.8)			
Occupation	Employed	38 (34.2)	15 (13.5)	58 (52.3)	16.35	0.288	0.000
	Self-employed	54 (61.4)	3 (3.4)	31 (35.2)			

**Table 4 ijerph-19-05606-t004:** Institution-related factors that affect the iCHF, according to healthcare providers and community members.

Institutional-Related Factors							
Ability to Pay the iCHF Premiums							
Factor	Categories	Agree	Neutral	Disagree	χ^2^	Cramer V	Sig.
Education Level	University	50 (37.0)	39 (28.9)	46 (34.1)	14.01	0.265	0.001
	Primary/Secondary	40 (62.5)	6 (9.4)	18 (28.1)			
Income Level	High	22 (30.6)	28 (38.9)	22 (30.6)	18.67	0.306	0.000
	Low	68 (53.5)	17 (28.7)	42 (33.1)			
Occupation	Employed	36 (32.4)	35 (35.1)	40 (36.0)	19.08	0.310	0.000
	Self-employed	54 (61.4)	10 (11.4)	24 (27.3)			
iCHF Membership Size							
Factor	Categories	Agree	Neutral	Disagree	χ^2^	Cramer V	Sig.
Education Level	University	50 (37.0)	31 (23.0)	54 (40.0)	20.329	0.320	0.000
	Primary/Secondary	19 (29.7)	1 (1.6)	44 (68.8)			
Income Level	High	27 (37.5)	22 (30.6)	23 (31.9)	21.818	0.331	0.000
	Low	42 (33.1)	10 (7.9)	75 (59.1)			
Occupation	Employed	44 (39.6)	27 (24.3)	40 (36.0)	21.289	0.327	0.000
	Self-employed	25 (28.4)	5 (5.7)	58 (65.9)			
Availability of Enrollment Officers for the iCHF Scheme							
Factor	Categories	Agree	Neutral	Disagree	χ^2^	Cramer V	Sig.
Education Level	University	56 (41.5)	34 (25.2)	45 (33.3)	46.026	0.481	0.000
	Primary/Secondary	59 (92.2)	1 (1.6)	4 (6.3)			
Income Level	High	21 (29.2)	24 (33.3)	27 (37.5)	25.439	0.358	0.000
	Low	21 (92.2)	15 (11.9)	29 (23.0)			
Occupation	Employed	82 (65.1)	37 (33.6)	44 (40.0)	67.748	0.585	0.000
	Self-employed	74 (45.8)	2 (2.3)	12 (13.6)			

## Data Availability

The data are available upon reasonable request from the corresponding author.

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
