# Peer review of "Factors Associated with the Implementation of an Improved Community Health Fund in the Ubungo Municipality Area, Dar es Salaam Region, Tanzania"

_ijerph, 2022, doi:10.3390/ijerph19095606_

Round 1

Reviewer 1 Report

This study on health management in places such as the Dar es Salaam region is very interesting, as it provides a descriptive study of the perceptions of the users who pay for health services.
The introduction is correct, the method used and the statistical analysis are well reflected, as are the results obtained.
The discussion offers interesting questions on the management of user fees and the limitations suggest further research of this topic.
In my opinion it should be published.
Best regards.

Author Response

We are very grateful for you insightful comments and observations.

Reviewer 2 Report

Since the article is "... a condensed form of the first author's master's degree thesis..." (line 331) may be a more detailed presentation of the questionnaire could help. 

Factors Associated with the Implementation of an Improved Community Health Fund in the Ubungo Municipality Area, Dar es Salaam Region, Tanzania by Goodluck Mselle, Peter Nsanya, Kennedy Diema Konlan,  Yuri Lee, Jongsoo Ryu, and Sunjoo Kang

Just key main points here:

  1. The goal of the article is not precisely defined: “This study analyzes health facilities, community factors, and governmental factors associated with the Improved Community Health Fund (iCHF) scheme Ubungo District, Tanzania” (lines 20-21). What this analysis is for?
  2. The study under review does not cover the government as suggested on line 87 but only healthcare providers and the community. Consequently, the research design should be adjusted.
  3. The size of the sample should be more clearly stated. “ A total of 13 iCHF members were selected from each facility to participate in the study. A simple random sampling method was used to select 16 healthcare facilities (…)}” (lines 103-105). This suggests that the size of the sample should be 208 (13 x 16)  while in Table 1 the total number of members is 174.  
  4. The results (Tables 3-5) are generalized leaving aside some interesting results. The latter refers, for example, to the disagreements of respondents with high incomes, employed workers, and university graduates concerning community behaviour/culture and the iCHF scheme (Table 4).
  5. The following sentence from Discussion: “The results of this study showed that aside from the education and income level of community members, the iCHF will perform better if healthcare facilities have sufficient medical supplies/equipment and healthcare providers” (lines: 243-245) does not have adequate support in the findings and possesses slightly speculative character.
  6. Strenghts and Limitations of the study (lines 292-295) cannot be reduced to just three sentences. Limitations should be presented in more detail.
  7. Conclusions do not show any value-added. To some extent, this is the result of the vaguely defined goal of the article (and the study).
  8. There are some discrepancies between the figures presented in the body of the article and the tables, for example, line 152: 12,57 vs. 12,56; line 166: 52,87 vs. 52,86, line 180: 46,03 vs. 46,03line 187: 12,08 vs. 12,07, line 207: 18,68 vs. 18,67, line 208: 19,09 vs. 19,08 . Check also for 21,329. 21,818, 21,289 and some others in Table 5  because in the body of the article it is a two digit number (line 204 and 212 for example).
  9. Table 3 must be modified - the content now under Table 4 on p. 6 must be included in the table.

Author Response

We are grateful for your insightful contributions in improving this manuscript

Reviewer 3 Report

This is an informative and sound study that may be useful to the contribution of knowledge in the area. While the authors have presented the study well, there are rooms for improvement:

  1. In the purpose of the study I would suggest: The study aims to illustrate or investigate... (the variables of the study), instead of to assist the government.
  2. It is better if the authors can give more information about the instrument, particularly what is measured in the health facility, government, and community related factors. It is not really clear when the result is presented.
  3. Do you have any reasons why the sample is purposive? Please explain and this may be also written in the limitation of the study.

Author Response

Thank you very much for your insightful contributions towards improving this manuscritp
